# Constrained-Differential-Kinematics-Decomposition-Based NMPC for Online Manipulator Control with Low Computational Costs

Jan Reinhold *[ID], Henry Baumann [ID] and Thomas Meurer [ID]

Automation and Control Group, Faculty of Engineering, Kiel University, Kaiserstraße 2, 24143 Kiel, Germany
* Correspondence: janr@tf.uni-kiel.de; Tel.: +49-431-880-6121

**Abstract:** Flexibility combined with the ability to consider external constraints comprises the main advantages of nonlinear model predictive control (NMPC). Applied as a motion controller, NMPC enables applications in varying and disturbed environments, but requires time-consuming computations. Hence, given the full nonlinear multi-DOF robot model, a delay-free execution providing short control horizons at appropriate prediction horizons for accurate motions is not applicable in common use. This contribution introduces an approach that analyzes and decomposes the differential kinematics similar to the inverse kinematics method to assign Cartesian boundary conditions to specific systems of equations during the model building, reducing the online computational costs. The resulting fully constrained NMPC realizes the translational obstacle avoidance during trajectory tracking using a reduced model considering both joint and Cartesian constraints coupled with a Jacobian transposed controller performing the end-effector's orientation correction. Apart from a safe distance from the obstacles, the presented approach does not lead to any limitations of the reachable workspace, and all degrees of freedom (DOFs) of the robot are used. The simulative evaluation in GAZEBO using the Stäubli TX2-90 commanded of ROS on a standard computer emphasizes the significantly lower online computational costs, accuracy analysis, and extended adaptability in obstacle avoidance, providing additional flexibility. An interpretation of the new concept is discussed for further use and extensions.

**Keywords:** kinematic analysis; robotic differential model decomposition; nonlinear model predictive control (NMPC); controller couplings; joint and Cartesian space constraints; computing time reduction; accuracy analysis; trajectory tracking; obstacle avoidance

## 1. Introduction

Modern industry is in a constant state of change driven by the contemporary labor market, the purchase demand, and the effective use of resources or machines [1]. Robots are increasingly being used in process automation to carry out monotonous and strenuous work, also reducing the operating costs [2,3]. In addition, efficient image recognition and sensor fusion enable increasingly accurate recognition of the environment in the robot's workspace [4]. Thus, using appropriate algorithms, robots can also be deployed in varying and disturbed environments to cover further fields of activity [5].

One sector undergoing a tremendous transformation is agriculture, which motivates this paper, but does not limit the scope of the presented approach. On the one hand, farmers, industry, and governments need to keep the costs moderate, even in high-wage countries, and on the other hand, consumers appreciate a sustainable and regional production [6]. These requirements are not mutually exclusive, but this is a subject area that needs to be developed, among other fields of application [7]. In particular, image recognition has been improved and adapted to specific agricultural problems in the last decade, allowing high-quality recognition with many features in widely disturbed environments [8–10]. Precision

agriculture enables, e.g., mechanical weed removal without damaging the adjacent plants, so that the use of pesticides can be reduced [11,12]. This usually requires equipment that is dedicated to a specific application and is expensive to purchase and maintain. In contrast, (industrial) robots are flexible and sustainable, as they are applicable for multiple applications throughout the agricultural season, simply by using different end-effectors. However, for the application in a distributed environment with multiple obstacles, trajectory planning and control have to be accomplished almost delay-free. Achieving low computational costs in optimal control using a robot with multiple degrees of freedom (DOFs) is addressed in this work.

Motion control is used for the adaption of planned trajectories in the Cartesian or joint space [13,14], which must subsequently be adjusted to a varying environment by means of a closed control loop. In general, either a discrete or a continuous interpretation of the workspace can be chosen. When choosing the discrete approach, the detected environment is meshed [15], and the optimal path is planned along the resulting nodes and edges [16]. Here, the Dijkstra and A* algorithms [17], as well as sampling-based methods can be applied with low computational cost to solve the planning task [18]. In general, inverse kinematics or Jacobian inverse controllers with low computational costs are subsequently used for the transformation into the joint space [19]. However, setting up the mesh is computational expensive, so that a delay-free motion control is not possible in a highly varying environment [20,21]. Examples of continuous motion planning tools include CHOMP, STOMP, and TrajOpt [22–24], which are commonly used. However, even though the computation times are short, they are not optimized to be used iteratively for delay-free control [25]. In addition, learning-based methods are increasingly applied, especially to take the aging of the robotic systems into account during motion control [26]. Using iterative learning control [27], motions are repeated until the solution is within an acceptable range. However, for motions in a varying environment, it is complicated to train these systems, as individual movements have to be run several times with the same initial and terminal states [28]. If a reference trajectory is known, the repetitive control approach can be added to be periodic and address the initialization problem [29]. Furthermore, reinforcement learning is used to improve the performance of the tracking controllers [30]. However, this paper presents a model-based control scheme that adapts the motion based on the robot's kinematic specifications. To realize a closed control loop, which iteratively considers varying environmental constraints in the Cartesian space and robotic constraints in the joint space given by the multi-DOF robot, nonlinear model predictive control (NMPC) is used [31]. As the dynamic optimization has to be solved on a receding horizon, computational efficiency is an issue for real-time application.

Two different time horizons have to be considered during the implementation of NMPC [32,33]. The prediction horizon specifies how far the movements in the disturbed environment are predicted. Governed by the sample size and the DOFs, the number of decision variables is set, which determines the computational costs to solve the optimal control problem (OCP) online [34]. Secondly, the control horizon, which is shorter than the prediction horizon, describes a kind of buffer along which the robot executes the movements of the last valid OCP solution [35]. A delay-free implementation of NMPC is not possible at the sampling rates of commonly used (robot) controllers, if an OCP for the prediction horizon were to be solved in every iteration step [36]. Thus, the control is maintained for subsequent samples along the control horizon, which is as accurate as the environment has been captured. Hence, faster solving of the OCP results in a shorter control horizon, and thus, rather optimal movements will be obtained [25]. A variety of approaches exist that perform NMPC [37–39], also involving horizon adaptions [40–42] and system reformulations [43,44]. However, in order to decrease the computational costs and, thus, the number of the decision variables, either the three-dimensional (3D) Cartesian space is only considered for the implementation of NMPC in robotics [45–47] or the number of actuated robot joints is reduced and particularly powerful hardware or software is used for the computations [25,48,49]. If only the Cartesian space is considered, the OCP neglects all

nonlinearities of the robot model and does not take the reachable work and joint space into account during the motion computations [50,51]. It must be ensured that the subsequent joint space transformation is reachable; otherwise, the OCP must be solved again with a different parameter set. Some approaches include the robotic constraints, but they limit the robots' DOFs to handle the computational expenses [52,53]. This complicates the general application of multi-DOF robots in disturbed environments, where all six Cartesian DOFs must be adjusted [54].

The approach introduced here analyzes the robot kinematics, thereby reducing the number of decision variables of NMPC to reduce the computational costs. It preserves the full robot workspace by adding an additional controller. Using kinematic analysis, this contribution addresses the cause of the computational costs themselves, rather than the symptoms, by means of adjustments in the implementation. Obstacle avoidance in 3D space is primarily performed by translational movements, i.e., evasion is achieved by displacement. In general, tilting the end-effector can also avoid collisions. However, the associated robotic joints provide a significantly smaller workspace, and simultaneously, the tool cannot perform the desired task in the correct orientation [55]. Referring to the agricultural context, manual weeding would have to be interrupted to avoid adjacent plants, which is less effective. The approach introduced here decomposes the differential kinematics analogously to the inverse kinematics method to partition the relevant equations and joints, respectively [56]. The procedure is applied to an industrial robot, which can be decomposed into the anthropomorphic arm and the spherical wrist, but it can be transferred to all multi-DOF robot types, which can also be separated into a translational and rotational part [57]. By splitting the problem, the constraints caused by external obstacles are assigned a priori and, therefore, do not need to be assigned during the online processing. As a result, two coupled controllers execute a constrained translational motion combined with a rotational movement for accurate trajectory tracking in a disturbed environment. The translational motion controller is realized as NMPC and considers both joint and Cartesian constraints. Compared to the consideration of the complete robot model, a significant reduction of the computational costs can be achieved due to the limited number of decision variables in the OCP. In this way, the consideration of additional boundary conditions to adjust the behavior of the robot still allows almost delay-free evaluations. Based on the joint control for the translational motion avoiding obstacles, a Jacobian transpose controller performs the rotation correction using the spherical wrist so that the end-effector maintains the correct alignment [58].

The proposed method is suitable for applications in various fields including industrial robots, where the dynamic parameters are typically unknown and can be realized even when using standard computer hardware due to the reduction of the computing times in the optimization. The approach can be used as well, e.g., for the online motion control of welding processes [59] or in the context of collaboration [60,61], which are common applications in industry requiring delay-free adaptation to a disturbed environment. For standard industrial robots, typically not only the dynamic parameters are unknown, but it is also advantageous to use only the kinematic specifications. In the case of model-based controllers, model uncertainties lead to performance losses in operation and inaccuracies due to the robot aging [62], for which the differential kinematics is taken into account in the presented approach. Here, the standard industrial robot Stäubli TX2-90 [63] is used as an example, which is simulated in Gazebo [64]. The communication is performed by means of ROS [65]. In the evaluation, the required computation times needed by the introduced NMPC approach are compared with the consideration of the full robot model in different setups. Further, the trajectory tracking accuracy is analyzed. Motivated by manual weed removal, a scenario is set up where the robot must adjust its initially given trajectory online to avoid damaging adjacent fixed and moving obstacles. Plants are abstracted as spheres so that objects recognized by image recognition, such as cabbages, can be easily integrated into a concrete application [66]. Weed removal itself is not shown, but the collision-free movements with correct alignment demonstrate the applicability of the approach [54],

which can be transferred to various industrial applications. Before the movement starts, a polynomial planned trajectory is specified, which crosses the obstacles. The end-effector must track this trajectory using NMPC and the Jacobian transpose controller. The NMPC detects the respective obstacles only within its prediction horizon, to which the movements must be adapted. The short evaluation times of the optimization allow the additional limitation of the achievable Cartesian workspace in height, which leads the translational motion to ground-level avoidance. In the context of the trajectory planning, an automatic selection of the joint configurations is presented, which replaces a manual selection, as is common for point-to-point movements in robotics, e.g., as utilized in [63,65,67,68].

The paper is organized as follows: Section 2 recalls the concept of the inverse kinematics for anthropomorphic robots and proceeds with the explanation of the differential kinematics decomposition. Based on this, the OCP for the end-effector's translational movement including joint and Cartesian constraints is described in Section 3, which is subsequently implemented as NMPC for online control. In addition, the Jacobian transpose controller for the rotation correction in 3D space is applied and coupled with the NMPC. In Section 4, the computation time savings of the approach, the trajectory tracking accuracy, and the control in a disturbed and varying environment including fixed and moving obstacles are demonstrated. A discussion of the results is provided in Section 5, and the final remarks in Section 6 conclude this contribution.

## 2. Modeling and Mathematical Decomposition of the Manipulator

For the decomposition of the robotic model, a standard industrial manipulator with $n$ revolute joints $q \in \mathbb{R}^n$ and an anthropomorphic structure consisting of an anthropomorphic arm and a spherical wrist was considered [69]; see Figure 1. A (non-)redundant robot with $n \geq 6$ was assumed, so that the workspace included all six Cartesian DOFs. Within the reachable workspace, an end-effector's pose is expressed by the homogeneous transformation matrix $H_w^e(q) \in SE(3)$, which comprises the translation vector $p_w^e(q) \in \mathbb{R}^3$ and the rotation matrix $R_w^e(q) \in SO(3)$. The subscript clarifies the reference system, while the superscript marks the body-fixed frame to be described therein. Thereby, the index $w$ represents the fixed world frame. Additionally, the end-effector frame $\{e\}$ describes the pose of the tool's point of interest mounted on the flange $\{f\}$. Based on the Denavit–Hartenberg (DH) convention [70], the direct kinematics of the robot can be derived.

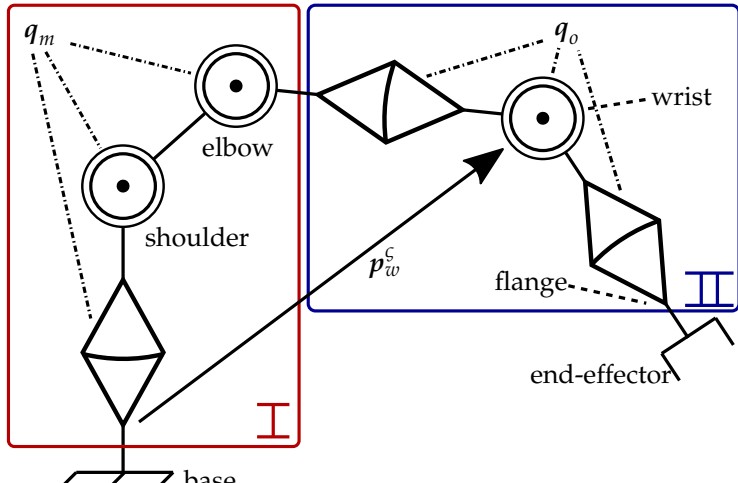

**Figure 1.** Manipulator with $n = 6$ revolute joints and an anthropomorphic structure consisting of an anthropomorphic arm (**I**) and a spherical wrist (**II**).

### 2.1. Analysis of the Inverse Kinematics

The method of inverse kinematic analysis [69] is recalled briefly in order to provide a better understanding and the motivation of the following sections. Inverse kinematics can be used to determine the associated joint angle configuration $q = [q_m^\mathsf{T}, q_o^\mathsf{T}]^\mathsf{T}$ given a desired

end-effector pose $H^e_{w,\text{des}}$. For the manipulator, which is shown exemplarily in Figure 1, the first $m$ joints $\boldsymbol{q}_m \in \mathbb{R}^m$ are assigned to the red framed anthropomorphic arm (I) and the last $o$ joints $\boldsymbol{q}_o \in \mathbb{R}^o$ belong to the blue framed spherical wrist (II). We considered $3 \leq m < n$ and $3 \leq o < n$ so that the $n = m + o$ robot DOFs are partitioned in such a way that any pose can be achieved in the reachable workspace. To this end, the robotic model is decomposed into the $\boldsymbol{q}_m$ associated part for translational displacement and the $\boldsymbol{q}_o$ relating system for rotational alignment to reduce the number of variables in the equations describing both associated models, respectively [21,71]. Thus, instead of evaluating the entire kinematic chain, two reduced chains are considered. The connection of the two systems is defined at the wrist frame $\{\varsigma\}$, where the so-called wrist point $\boldsymbol{p}^\varsigma_w = [p^\varsigma_{w,x}, p^\varsigma_{w,y}, p^\varsigma_{w,z}]^\text{T} \in \mathbb{R}^3$ is located. The wrist point can be obtained by equating the traversed kinematic chains that converge in $\{\varsigma\}$ starting in the $\{w\}$ and the $\{e\}$ frame, respectively. Starting at $\{e\}$, the desired end-effector pose can be projected onto the flange by

$$H^f_w = \begin{bmatrix} R^f_w & \boldsymbol{p}^f_w \\ \mathbf{0}^\text{T} & 1 \end{bmatrix} = H^e_{w,\text{des}} \left(H^e_f\right)^{-1} = H^e_{w,\text{des}}\, H^f_e \tag{1}$$

using the tool-specific transformation matrix $H^e_f$.

The orientation of the $z_f$-axis of the flange is denoted by $\boldsymbol{r}^f_{w,z}$ and can be obtained from the third column of $R^f_w$. The wrist point

$$\boldsymbol{p}^\varsigma_w = \boldsymbol{p}^f_w - d_f\, \boldsymbol{r}^f_{w,z} \in \mathbb{R}^3 \tag{2}$$

can be calculated based on (1) and the length $d_f$ of the last link ending at the flange. Using (2) as the left-hand side and the position vector of the direct kinematics $H^\varsigma_w(\boldsymbol{q}_m)$ as the right-hand side, a system of equations can be set up to determine $\boldsymbol{q}_m$. When considering a redundant robot with $m > 3$ and $n > 6$, additional constraints to the nullspace must be introduced to solve the system uniquely [72]. Apart from the possible nullspace, there are generally up to four valid solutions describing the shoulder left or right and elbow up or down configurations [73]. The so-called rotation correction can be determined by

$$R^f_m = \left(R^m_w\right)^\text{T} R^f_w = R^w_m R^f_w. \tag{3}$$

Via the $ZYZ$-sequence [74], which is based on the joint structure of the spherical wrist, $R^f_m$ can be implied for the joints $\boldsymbol{q}_o$. This in turn yields ambiguous solutions known as wrist top and wrist bottom [73], respectively. This doubles the maximum number of possible configurations mentioned above, so that up to eight solutions can exist for one desired pose $H^e_{w,\text{des}}$. In this contribution, an automatic selection was introduced, which performs an evaluation of the most-suitable joint configuration and selects it for the movement. Jumps between the up to eight solutions are avoided, and the common boundaries are considered.

### 2.2. Decomposition of Differential Kinematics

Based on the method presented before, the robot model is decomposed for the following control architecture. The separation into a translational and a rotational part allows the consideration of external boundary conditions, e.g., for the avoidance of obstacles, to be directly assigned to specific joints in the robot's kinematic chain. Thus, the DOFs considered in the optimization-based control approach are reduced by kinematic analysis, reducing the computational costs. Differential kinematics rather than direct kinematics was taken into account to avoid algebraic loops in the online computations [75] and for a more straightforward restriction of the nullspace in the case of redundant robots [57,69].

The transformation of joint velocities $\dot{q} \in \mathbb{R}^n$ into Cartesian velocities can be performed by differential kinematics using

$$\begin{bmatrix} \dot{p}^e_w \\ \omega^e_w \end{bmatrix} = J^e_w(q)\, \dot{q} \in \mathbb{R}^6. \tag{4}$$

The translational velocity of the end-effector with respect to the $\{w\}$ system is described by $\dot{p}^e_w \in \mathbb{R}^3$, while $\omega^e_w \in \mathbb{R}^3$ expresses the corresponding angular velocities [69]. The nonlinear geometric Jacobian matrix:

$$J^e_w(q) = \begin{bmatrix} J_{\text{trans}} \\ J_{\text{rot}} \end{bmatrix} = \begin{bmatrix} J_{\text{trans},1} & J_{\text{trans},2} \\ J_{\text{rot},1} & J_{\text{rot},2} \end{bmatrix} \in \mathbb{R}^{6 \times n} \tag{5}$$

is introduced. From (5), it can be seen that the entire Jacobian consists of a translational and a rotational part. Transferring the approach of Section 2.1 to the decomposition of the manipulator, $J_{\text{trans},1}$ and $J_{\text{rot},2}$ represent the associated terms in the differential kinematics, and the submatrices $J_{\text{trans},2}$ and $J_{\text{rot},1}$ denote cross-couplings, respectively. Instead of using the entire transformation from (4), the differential kinematics is split as well. Based on the decomposition analyzed in the inverse kinematics, translation is performed by the first $m$ and orientation by the last $o$ robot joints [57]. According to the general matrix computation as described in [69], the matrices $J^\varsigma_{w,\text{trans}}(q_m) \in \mathbb{R}^{3 \times m}$ and $J^f_{m,\text{rot}}(q_o) \in \mathbb{R}^{3 \times o}$ are introduced and used subsequently instead of (5). The DOFs due to the cross-couplings are eliminated as a consequence in the transformation performed in (4) for the full robot system. This means that the joints $q_m$ no longer have an active influence on the end-effector's orientation and $q_o$ cannot be used for the translational positioning of the $\{e\}$ frame. Further, two controllers for $q_m$ and $q_o$ were designed separately and then coupled.

One controller controls the positioning using $J^\varsigma_{w,\text{trans}}(q_m)$, and the other controller adjusts the alignment with $J^f_{m,\text{rot}}(q_o)$. Analogous to the evaluation of inverse kinematics in Section 2.1, there is no loss of DOFs in the Cartesian space, and due to controller couplings, the entire workspace is still reachable.

It should be emphasized that the translational part $J^\varsigma_{w,\text{trans}}(q_m)$ refers to the wrist point $p^\varsigma_w$, while the orientation of the $\{\varsigma\}$ system is irrelevant in this context. Using (1) and (2), the desired wrist point is obtained from $p^e_{w,\text{des}}$, and an orientation error follows from the wrist positioning using the first $m$ robot joints. In turn, the Jacobian $J^f_{m,\text{rot}}(q_o)$ for the rotational part refers to the $\{m\}$ system localized in the robot's elbow, the last joint of the anthropomorphic arm, as shown in Figure 1. The link between the $\{m\}$ and the $\{\varsigma\}$ system exhibits a constant length and is aligned along the rotation axis of the first spherical wrist joint. Thus, this DOF only changes the alignment and not the displacement between both systems, and the two kinematic chains can be connected in this way.

## 3. Optimal Trajectory Control Using Decomposed Differential Kinematics

To implement fast online control, Section 3.1 presents a computationally effective planning scheme involving all robot joints to generate an initial trajectory that does not take external disturbances into account. It can be used when the workspace is not constrained and serves as a reference in a warm start of the following optimization. An automatic selection is introduced that identifies the most-suitable joint configuration for the desired terminal pose. The up to eight possible solutions of the inverse kinematics are checked for jumps for the planned pose transition, and the solution with the largest distance to the joint boundaries is selected. In Section 3.2, the constrained OCP is formulated, with the translational part of the decomposed system from Section 2.2 as the underlying model. The OCP is evaluated on a receding time horizon, i.e., controlling the robot using NMPC. Meanwhile, the orientation of the end-effector is considered separately using the controller presented in Section 3.3. In Table 1, the main difficulties and characteristics of the two controllers are listed as an overview. Special attention has to be paid to the wrist position, which is iteratively placed by the NMPC and determines the starting pose for the

orientation controller. The combination of the two controllers provides the overall control of the robot, and both are calculated subsequently in each iteration. If the orientation controller, based on the Jacobian transpose controller here, is also implemented as a second NMPC, the controllers would have to be evaluated sequentially and, therefore, would be time consuming because of the dependency with respect to the reference pose at the $\{m\}$ frame.

**Table 1.** Comparison of the difficulties and characteristics of the decomposed robot model illustrated in Figure 1 for optimal trajectory control achieving low computation times in online calculations.

| Properties | I: Anthropomorphic arm | II: Spherical Wrist |
|---|---|---|
| intended use | • translational movement<br>• avoiding obstacles in disturbed environment | • orientation control<br>• alignment for desired rotation |
| constraints | • iterative solving starting at fixed base<br>• environmental, state and input constraints<br>• consideration of the distance between $p_w^{\varsigma}$ and end-effector | • depending on wrist movement<br>• standard controller bounded to limits<br>• compensation of rotation correction |
| control | • control of $q_m$ using NMPC<br>• optimization with known reference | • control of $q_o$ with Jacobian controller<br>• reaction based on wrist displacement (from NMPC) |
| singularity avoidance | • configuration bounded to objective function with regularization<br>• preselection of the closest solution | • Jacobian transpose using no inversion in calculations<br>• unit quaternions preventing a *Gimbal lock* [74] |

**Remark 1.** *Here, only a multi-DOF robot with an anthropomorphic structure and revolute joints is discussed, so that an independent assignment of the joints to a translational and rotational motion in the Cartesian space can be performed. This design as an open or closed kinematic chain is the most common setup of standard industrial robots. A transfer of the approach to other manipulator types can be applied if the robotic model admits a decomposition according to the specification.*

*3.1. Polynomial Trajectory Planning*

An initial planning for all $n$ robot joints is performed before the online controlled robot movements start. To generate a reference trajectory, a polynomial approach in the joint space is utilized to connect the initial end-effector pose represented by the homogeneous transformation $H_w^e(q(t_0))$ at time $t_0 \in \mathbb{R}_{\geq 0}$ with the desired terminal pose $H_{w,\text{des}}^e$ at time $t_1 = t_0 + T$, obtaining a continuously differentiable trajectory. The transition time $T \in \mathbb{R}_{>0}$ must be chosen so that the dynamic joint limits of the robot are not violated. To check whether $H_{w,\text{des}}^e$ is an admissible pose with the mounted end-effector according to the given bounds in the robot's data sheet, (2) can be used to validate the wrist point. As mentioned in Section 2.1, up to eight possible joint configurations can be determined for the given pose at $t_1$. From the set of possible solutions of the inverse kinematics, the configurations that are not included in the reachable joint space:

$$\mathcal{Q} := \{q \in \mathbb{R}^n \mid q_{\min} \leq q \leq q_{\max}\} \tag{6}$$

are excluded. To connect the initial joint setup $q(t_0)$ and the remaining $\beta \leq 8$ terminal configurations $Q_\beta(t_1) = [q_1(t_1), \ldots, q_\beta(t_1)] \in \mathbb{R}^{n \times \beta}$, the polynomial:

$$\gamma(t) = \sum_{j=0}^{7} \lambda_j \, t^j \in \mathbb{R}^n \tag{7}$$

is introduced. The eight unknowns $\lambda_j$, $j \in \{0, \ldots, 7\}$ for each of the $n$ joints can be determined, respectively, from the eight boundary conditions:

$$\gamma_i^{(j)}(t_0) = q^{(j)}(t_0), \quad \gamma_i^{(j)}(t_1) = q_i^{(j)}(t_1), \quad j \in \{0, \ldots, 3\}, \; i \in \{1, \ldots, \beta\} \tag{8}$$

for each configuration $i$. In (8), $\gamma_i(t) = \gamma_i^{(0)}(t)$ applies, and the derivatives are given by $\gamma_i^{(j)}(t)$, which describe the associated velocity, acceleration, and jerk, respectively. The velocity bounds can be measured or formed by the inverse evaluation of (4). Without loss of generality, the acceleration and jerk are chosen to be zero at the beginning and at the end of the transition. The acceleration bounds can alternatively be transformed by introducing the time derivative of the Jacobian in (5) [69]. In order to drop the solutions that contain an unnecessary change at the shoulder, elbow, or wrist of the robot, all $\beta$ transitions connecting $q(t_0)$ with the configurations in $Q_\beta(t_1)$ are sampled by $t_\gamma \in \mathbb{R}_{>0}$ and examined for jumps. From the remaining $r \leq \beta$ possibilities that do not involve jumps, the configurations that are furthest away from the joint boundaries with the corresponding joints are selected from

$$\max_{\rho} \Big\{ \min_{i} \{ q_{i,\rho}(t_\gamma) - q_{i,\min}, q_{i,\max} - q_{i,\rho}(t_\gamma) \} \Big\}, \quad \rho \in \{1, \ldots, r\}, \; i \in \{1, \ldots, n\}. \tag{9}$$

Each joint $i$ is evaluated individually at each sample step $t_\gamma$. If several joint configurations exhibit the same distances to the bounds, the maximum operator in (9) is used to select the configuration that maintains the greatest distance from the boundaries $q_{\min}$ and $q_{\max}$, considering all sampling steps $t_\gamma$. If the coupling of the two checks were reversed, a joint that is far from its bound could compensate a joint close to its respective bound in the evaluation. Since the planning is implemented in the joint space, no consideration of the *Euler angles* [74] in Cartesian space is necessary, which prevents representation singularities. Using the introduced procedure in (9), an automatic selection method of the most appropriate kinematic configuration is introduced, eliminating the need for a manual selection, required by most of the inverse kinematics tools [63,65,67,68].

### 3.2. Optimization-Based Translational Trajectory Control

For the translational motion in the robot's workspace considering obstacles, a constrained optimization problem with a fixed end time $\tau_1 = \tau_0 + N \in \mathbb{R}_{>0}$ is introduced. The prediction horizon of the OCP is defined by $N \in \mathbb{R}_{>0}$ and is starting at $\tau_0 = t_\delta$. Thereby, $t_\delta \in \mathbb{R}_{\geq 0}$ describes the current sampling step. With the underlying model of the decomposed differential kinematics from Section 2.2, the joint velocities:

$$\mathcal{U} := \{ u \in \mathbb{R}^m \mid -\dot{q}_{m,\max} \leq u(\tau) \leq \dot{q}_{m,\max} \} \tag{10a}$$

are chosen as fictitious inputs of the OCP. As can be seen from (10a), only the first $m$ joints of the robot are taken into account for the displacement of the wrist point $p_w^\varsigma$. The end-effector orientation is adjusted subsequently by means of $q_o$. Standard industrial robots are usually controlled using joint position controllers [63,65] so that the joint angles $q_m$ to command the translational motion of the robot can be obtained by solving $\dot{q}_m = u$. Furthermore, the OCP:

$$\min_{u \in \mathcal{U}} F(u) = \int_{\tau_0}^{\tau_1} l\big(u, p_w^\varsigma(q_m), \ddot{q}_m, \mu\big) \, \mathrm{d}\tau \tag{10b}$$

is considered by minimizing the running cost:

$$l\left(\boldsymbol{u}, \boldsymbol{p}_w^\varsigma(\boldsymbol{q}_m), \ddot{\boldsymbol{q}}_m, \boldsymbol{\mu}\right) = \mu_u\, \boldsymbol{u}^{\mathrm{T}}\boldsymbol{u} + \mu_{\ddot{q}}\,\ddot{\boldsymbol{q}}_m^{\mathrm{T}}\,\ddot{\boldsymbol{q}}_m + \mu_p\left(\boldsymbol{p}_{w,\mathrm{des}}^\varsigma - \boldsymbol{p}_w^\varsigma(\boldsymbol{q}_m)\right)^{\mathrm{T}}\left(\boldsymbol{p}_{w,\mathrm{des}}^\varsigma - \boldsymbol{p}_w^\varsigma(\boldsymbol{q}_m)\right) \quad (10\mathrm{c})$$

over the time interval $\tau \in [\tau_0, \tau_1]$. The elements from $\boldsymbol{\mu} = [\mu_u,\ \mu_{\ddot{q}},\ \mu_p]^{\mathrm{T}} \in \mathbb{R}^3$ can be used to weight the individual terms in (10c). The desired wrist position $\boldsymbol{p}_{w,\mathrm{des}}^\varsigma$ is derived from the desired end-effector pose using (2). If a reference trajectory is specified, e.g., with the procedure introduced in Section 3.1, using MoveIt for task-level motion planning [76] or based on the techniques summarized in [14], the Lagrange function shown in (10c) aims for trajectory tracking. Alternatively, only the terminal position could be passed to (10c) as a reference, which is called a local motion planning problem [35]. By considering the input $\boldsymbol{u}$ in (10c), the agility can be influenced, and the relating part also represents a regularization term, to prevent singular arcs [43]. In order to further prevent singularities, it is possible to include the manipulability measure into the running cost as well [77]. In practice, the integral listed in (10b) is discretized by a sum over $k \in \mathbb{N}_{>0}$ temporal grid points for the numerical implementation. Enabling an influence on the rate change, $\ddot{\boldsymbol{q}}_m$ is also included in (10c). The acceleration $\ddot{\boldsymbol{q}}_m$ and the jerk $\dddot{\boldsymbol{q}}_m$ result from the discrete differentiation of the input $\boldsymbol{u}$, respectively. Based on the system formulation:

$$\frac{\mathrm{d}}{\mathrm{d}\tau}\begin{bmatrix}\boldsymbol{p}_w^\varsigma \\ \boldsymbol{q}_m\end{bmatrix} = \begin{bmatrix}J_{w,\mathrm{trans}}^\varsigma(\boldsymbol{q}_m)\,\boldsymbol{u} \\ \boldsymbol{u}\end{bmatrix},\ \tau > \tau_0 \quad \text{with}\ \begin{bmatrix}\boldsymbol{p}_w^\varsigma(\tau_0) \\ \boldsymbol{q}_m(\tau_0)\end{bmatrix} = \begin{bmatrix}\boldsymbol{p}_{w,0}^\varsigma \\ \boldsymbol{q}_{m,0}\end{bmatrix}, \qquad (10\mathrm{d})$$

the variables in (10b) can be determined. Here, $\boldsymbol{p}_{w,0}^\varsigma$ describes the initial wrist position and $\boldsymbol{q}_{m,0}$ the initial joint angles at time $\tau_0$. Using the inequality constraints:

$$\begin{aligned}\boldsymbol{q}_{m,\mathrm{min}} \leq\ &\boldsymbol{q}_m\ \leq \boldsymbol{q}_{m,\mathrm{max}} \\ -\boldsymbol{q}_{m,\mathrm{max}}^{(j)} \leq\ &\boldsymbol{q}_m^{(j)} \leq \boldsymbol{q}_{m,\mathrm{max}}^{(j)},\quad j \in \{2,3\},\end{aligned} \qquad (10\mathrm{e})$$

the system (10d) is constrained to the reachable joint space, since the selected joint angles $\boldsymbol{q}_m$ must be inside the valid bounds of (6). Applying the constraints in (10e) to the acceleration $\ddot{\boldsymbol{q}}_m$ and jerk $\dddot{\boldsymbol{q}}_m$, non-adjustable changes can be avoided. The respective bounds are usually known for standard industrial robots and can be taken from the appropriate data sheet, e.g., given by [63].

Obstacles are subsequently modeled as spheres to illustrate the approach [78], but can also be described by using sophisticated techniques as, e.g., by the evaluation of tetrahedral meshes or polyhedra [21,79,80]. Let $\nu$ denote the number of obstacles in the Cartesian space. These are assumed moveable and centered at $\boldsymbol{p}_{w,i}^\nu(\tau) \in \mathbb{R}^3$, $i \in \{1,\ldots,\nu\}$, imposing the inequality constraints:

$$r_i + d_f + \sqrt{(a_e)^2 + (d_e)^2} < |\boldsymbol{p}_w^\varsigma - \boldsymbol{p}_{w,i}^\nu(\tau)|, \quad \forall i = 1,\ldots,\nu. \qquad (10\mathrm{f})$$

Due to the decomposition of the robot model, the resulting orientation of the end-effector during motion is not known in the optimization. Therefore, the length of the spherical wrist plus the DH parameters $a_e \in \mathbb{R}$ and $d_e \in \mathbb{R}$ of the end-effector are also defined as a sphere around the wrist point. This is added to the radius $r_i \in \mathbb{R}_{>0}$ of each obstacle to obtain a safe distance.

For example, to perform horizontal motion only or to prevent touching the ground, the height:

$$p_{w,z,\mathrm{min}}^\varsigma\ \leq\ p_{w,z}^\varsigma\ \leq p_{w,z,\mathrm{max}}^\varsigma \qquad (10\mathrm{g})$$

of the robot's workspace can also be optionally bounded. Constraining of the OCP (10) by adding (10g) usually increases the computational times significantly, which will be demonstrated in Section 4.1. However, by reducing the robot model in (10d), it is possible to include further constraints influencing the robot's behavior.

NMPC can be applied by solving the presented OCP on a receding horizon with a suitable prediction length $N$. Both the initial wrist position and the joint angles can be obtained from the measured joint angles of the robot. Direct multiple shooting is used for the numerical evaluations, which considers an initial value problem in each time interval $[\tau_{\kappa-1}, \tau_\kappa]$, $\kappa \in \{1, \ldots, k\}$ [81]. Hence, $k$ initial value problems have to be solved in total. Since the subintervals can be solved independently, parallelization can be used. To ensure a continuous transition between intervals, boundary conditions must be imposed so that the boundary values of the adjacent intervals are identical [82]. For a warm start, the initial estimates of the optimization variables and the input can be set for each step $\tau_\kappa$ by the procedure presented in Section 3.1. According to [81], an approach is used here that first discretizes and then optimizes, converging to local or global minima depending on the solver settings and the weightings chosen in the quadratic objective function (10b).

### 3.3. Jacobian Transpose Controller Achieving Desired Orientation

To achieve the desired orientation, a controller is presented to track the last joints $q_o$ of the robot's kinematic chain accordingly. For this purpose, the Jacobian transpose controller is used and applied to the problem formulation [57]. This implies less computational effort compared to the Jacobian inverse controller commonly used in robotics and can be utilized to cross kinematic singularities [69]. When solving the NMPC formulated in Section 3.2, a joint configuration $q_m(t_\delta)$ for the first $m$ robot joints of the kinematic chain is obtained for each iteration step. These define the orientation of the $\{m\}$ frame at the robot's elbow, which results in the rotation matrix $R_w^m(q_m)$. This matrix can be used in (3) to determine the deviation matrix $R_m^f(q_m)$ between the desired end-effector orientation, transformed to the flange $\{f\}$ and the current wrist orientation governed by $q_m$. Therefore, the corresponding unit quaternions $[\eta_m^f(q_m), (\zeta_m^f(q_m))^T]^T$ can be derived [83,84]. They specify the desired unit quaternions with respect to the $\{m\}$ frame depending on the displacement of the $\{\varsigma\}$ system at the wrist point performed by the NMPC. From the joint angles $q_o(t_\delta)$ at sampling step $t_\delta$, the current $[\eta_m^f(q_o), (\zeta_m^f(q_o))^T]^T$ unit quaternions can be calculated. The orientation error:

$$\tilde{e}_m^f = \eta_m^f(q_o)\,\zeta_m^f(q_m) - \eta_m^f(q_m)\,\zeta_m^f(q_o) - S\big(\zeta_m^f(q_m)\big)\,\zeta_m^f(q_o) \in \mathbb{R}^3 \tag{11}$$

between these unit quaternions can be inferred, where the skew symmetric operator [85] reads

$$S(s_1, s_2, s_3) = \begin{bmatrix} 0 & -s_3 & s_2 \\ s_3 & 0 & -s_1 \\ -s_2 & s_1 & 0 \end{bmatrix} \in \mathbb{R}^{3\times 3}. \tag{12}$$

It should be emphasized that $\eta = 1$ holds true for the real part of the unit quaternions when the orientation is aligned, and thus, the orientation error in (11) can be expressed as a 3D quantity [86]. Using

$$\dot{q}_o = \big(J_{m,\text{rot}}^f\big)^T K \tilde{e}_m^f, \tag{13}$$

the feedback is imposed, including the positive definite matrix $K \in \mathbb{R}^{o\times 3}$ and the Jacobian determined in Section 2.2. The weighting matrix $K$ is bounded to the sample time and influences the speed of convergence. The required joint angles $q_o$ to control the robot are obtained by the integration of (13). In order to analyze the stability of the orientation controller, the Lyapunov function:

$$V = \big(\eta_m^f(q_m) - \eta_m^f(q_o)\big)^2 + \big(\zeta_m^f(q_m) - \zeta_m^f(q_o)\big)^T \big(\zeta_m^f(q_m) - \zeta_m^f(q_o)\big) \tag{14a}$$

is considered. After substituting the propagation equations for quaternions [86] into the rate of change:

$$\dot{V} = -\big(\tilde{e}_m^f\big)^T K \tilde{e}_m^f \tag{14b}$$

of (14a), the asymptotic stability of the orientation controller can be concluded. Thus, the controller converges to the desired orientation and is able to cross singularities, whereas, in contrast to the Jacobian inverse, it may deviate during the transition phase [58,69].

## 4. Simulation Results and Evaluation

To show that standard computer hardware is sufficient for the online calculations of the introduced NMPC for the translational motion including Cartesian and joint constraints coupled with the Jacobian transpose controller for the orientation correction, a standard computer with 16GB RAM and an Intel Core i7-8550U processor running Linux Ubuntu 18.04 was utilized. The iterative solution of the OCP is calculated using the MATLAB-interface [87] from CASADI [88] with the interior-point (IPOPT) algorithm [89]. As can be seen in Figure 2, the standard industrial 6-DOF robot Stäubli TX2-90 with a black rod as the end-effector and the DH parameters from Table 2 was used to present the introduced approach. Based on the decomposition performed in Section 2.2, both controllers consider different kinematic chains, respectively. Thus, the associated DH parameters to the $\{m\}$ system and the wrist point $p_w^\varsigma$ are also listed in the table. The $n = 6$ revolute joints are equally partitioned with $m = 3$ and $o = 3$ for the translational and rotational controllers.

**Table 2.** Denavit–Hartenberg (DH) parameters of the considered 6-DOF industrial Stäubli TX2-90 manipulator.

| $i$ | $a_i$ (mm) | $\alpha_i$ (rad) | $d_i$ (mm) | $\theta_i$ (rad) |
|---|---|---|---|---|
| $w$ | 0 | 0 | $-478$ | 0 |
| 1 | 50 | $-\pi/2$ | 478 | $q_1$ |
| 2 | 425 | 0 | 0 | $q_2 - \pi/2$ |
| $3/m$ | 0 | $\pi/2$ | 50 | $q_3 + \pi/2$ |
| $\varsigma$ | 425 | 0 | 50 | $q_3$ |
| 4 | 0 | $-\pi/2$ | 425 | $q_4$ |
| 5 | 0 | $\pi/2$ | 0 | $q_5$ |
| $f$ | 0 | 0 | 100 | $q_6$ |
| $e$ | 0 | 0 | 150 | 0 |

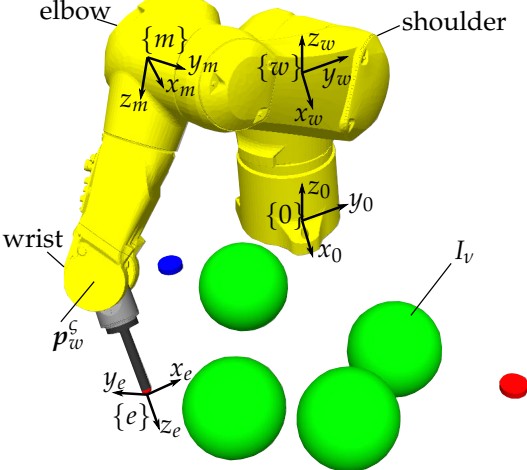

**Figure 2.** Standard industrial 6-DOF manipulator Stäubli TX2-90 [63] with an end-effector, visualized in GAZEBO [64,90]. The exemplary movement starts at the blue marker and ends at the red one. The green obstacles are only considered in Sections 4.3 and 4.4, but are not present in Section 4.2.

The evaluation consisted of four different demonstrations to highlight the performance of the NMPC approach based on the decomposed robot model. In Section 4.1, the

computation times of the introduced approach are evaluated and compared to an NMPC considering the full robot system. This highlights the significant difference in the computational costs between those approaches. Despite the decomposition of the model and the controller couplings, no losses in the applicability by the approach occur, which is shown in the following three evaluations. The trajectory tracking accuracy in the undisturbed case is discussed in Section 4.2 to show that the method can be used as an online motion controller. Subsequently, in Section 4.3, obstacles are placed in the environment. In the presented scenario, the end-effector must be guided between them without causing collisions. The obstacles are not taken into account for the trajectory planning described in Section 3.1, but will be avoided by the online controller introduced in Section 3.2, which perceives them only within its predictive horizon. Finally, in Section 4.4, moving obstacles are considered and the collision-free guidance of the end-effector in this setup is investigated.

### 4.1. Quantitative Analysis of the Computation Times

In the analysis of the reduction in the computation times, the presented approach was compared to an NMPC that considers the full robotic model. Using the Stäubli TX2-90, the NMPC based on the full system utilizes $n = 6$ joints as decision variables in each optimization step and incorporates both translation and orientation by $J_w^e(\boldsymbol{q})$ from (5). On the contrary, the decomposed system requires only $m = 3$ decision variables in its OCP and controls the orientation in parallel with the remaining three DOFs using the coupled Jacobian transpose controller. It should be emphasized that (10c) must be extended in the NMPC of the full system to include orientation as well. The objective functions of the two systems differ, but each was designed for the quantitative comparison. As listed in Table 3, three different scenarios consisting of no obstacle, one obstacle, and one obstacle including height constraints were compared to analyze the computation times. Additionally, two different prediction horizons $N_1 = 100$ ms and $N_2 = 200$ ms partitioned with $k_1 = 10$ and $k_2 = 20$ grid points were considered, achieving discrete intervals with a length of 10 ms, respectively. The discretization of the inputs to be determined corresponds to the update rate of the robot controller. Various converging point-to-point (PTP) movements covering the entire workspace of the robot were run multiple times, and the average computation time $\bar{t}$ per optimization step was recorded. This time and the standard deviation $\sigma$, which expresses the fluctuation of $\bar{t}$ required for one OCP, denotes the online capability of the NMPC. Note that the optimization was carried out until an optimal solution was found, but could be further shortened by limiting the maximum iterations, as done in [91]. Here, the stop condition for the objective function (10b) was set to a tolerance of $10^{-8}$, and if below this limit, the value did not change more than $10^{-6}$, indicating a minimum.

**Table 3.** Comparison of the averaged computation times $\bar{t}$ with the standard deviation $\sigma$ per optimization step of the respective NMPC.

| Point-to-Point Movement | | Decomp. System $\bar{t} \pm \sigma$ (ms) | Full System $\bar{t} \pm \sigma$ (ms) |
|---|---|---|---|
| without obstacles | $N_1$ | $26 \pm 1$ | $130 \pm 12$ |
| | $N_2$ | $30 \pm 2$ | $216 \pm 21$ |
| with obstacle | $N_1$ | $27 \pm 2$ | $153 \pm 53$ |
| | $N_2$ | $33 \pm 2$ | $209 \pm 9$ |
| with obstacle and height constraints | $N_1$ | $29 \pm 2$ | $230 \pm 136$ |
| | $N_2$ | $36 \pm 6$ | $284 \pm 112$ |

From Table 3, the comparison between the decomposed and the full system shows that the decomposed system requires only 10 % to 20 % of $\bar{t}$ to achieve an optimal solution and possesses lower deviations $\sigma$, independent of the scenario or the prediction horizon. Both $\bar{t}$ and $\sigma$ are important factors to be considered using NMPC in varying environments.

The Jacobian transpose controller evaluates the orientation error in each iteration and is only constrained to the gain matrix $K$.

It can be seen that the average computation times $\bar{t}$ for the full system increased significantly with the complexity of the scenario. This effect did not occur with the decomposed model, as the full system involves more nonlinearities, which must be taken into account to solve the OCP. On the one hand, higher computational costs yield lower possible update rates of the NMPC, which restricts the ability to act in rapidly varying environments. On the other hand, it is evident from Table 3 that $N_2$ increased the computation times for the full system by up to 67 %, in contrast to $\bar{t}$, when $N_1$ was chosen. This means that the choice of the prediction horizon limits the online capabilities. Using the decomposed robot model approach, the evaluation times increased by a maximum of 24 % using $N_2$ instead of $N_1$. Thus, the comparison of both the absolute and the relative computation times revealed a significantly higher performance of the presented decomposition-based method.

A second important factor is $\sigma$, which is a measure of the reliability to achieve $\bar{t}$. Smaller standard deviations, even between different scenarios, indicate that $\bar{t}$ is more likely to be achieved. In contrast to the NMPC considering the full system, the lower $\sigma$ of the decomposed system also allows for easier applicability to different tasks, since the NMPC does not converge for an unexpectedly long time in a more complex scenario. The choice of the control horizon is determined by the length of expected calculation times and should be kept as short as possible. Referring to Table 3, the control horizon can be set more reliably using the decomposed approach. The NMPC controller remains capable in online operation without delaying the robot's motion, resulting in $\bar{t}$ being larger than the set control horizon in the implementation.

### 4.2. Trajectory Tracking Accuracy of the Controller

In Section 4.1, the significant reduction of the computational costs is presented. Furthermore, it is shown that this did not lead to any restrictions in the motion behavior of the robot. The simulative setup for evaluating the introduced approach involving the NMPC and the orientation controller was built in GAZEBO [64]. The joint position controlled robot shown in Figure 2 is commanded by means of ROS [65,90]. The prediction horizon was chosen as $N = 100$ ms with $k = 10$ grid points per iteration, while the control horizon involved four discretization steps of 10 ms each. This means that, for all four consecutive updates of the robot commands, the solution from the buffer was used before being updated. Considering the average computation times from the previous subsection, this allows for online calculations without delaying the robot's motion due to the too long computations solving the OCP. Finally, the gain matrix for the orientation control was set to $K = \mathrm{diag}(20, 20, 20)$, and the weights of the NMPC's running costs in (10c) were chosen to be $\mu = [10^4, \, 10^2, \, 10^4]^\mathrm{T}$.

We omit the comparison with the full system in the following evaluations, on the one hand, for the sake of readability and, on the other hand, to avoid having an unfair comparison realized. As shown in Section 4.1, no delay-free execution can be realized for the chosen $N$ using the full system, which distorts the comparison. Depending on the controller settings, we observed only minor to no deviations between the results in internal comparisons, depending on the scenario.

As shown in Figure 2, the end-effector has to move from the blue marker with $p^e_{w,0} = [110, \, -350, \, -405]^\mathrm{T}$mm at $t_0$ to the red marker with $p^e_{w,\mathrm{des}} = [780, \, 390, \, -405]^\mathrm{T}$mm at $t_1$. None of the green obstacles are considered in this subsection when performing the trajectory tracking analysis, and they are only drawn in preparation for the next scenario. The desired orientation was set to $R^e_{w,\mathrm{des}} = \mathrm{diag}(-1, 1, -1)$, meaning that the end-effector has to point vertically downwards. However, all other orientations reachable in the manipulator's workspace can also be realized. As explained in Section 3.3, the Jacobian transpose controller is asymptotically stable and does not induce singularities in individual joint configurations, e.g., compared to the Jacobian inverse controller. As illustrated in Figure 3b with dashed lines, the set point change of the desired position using the polyno-

mial approach from Section 3.1 starts at $t_0 = 0.5\,\mathrm{s}$ and ends at $t_1 = 2\,\mathrm{s}$. The demonstration scenario involving a short transition time $T = 1.5\,\mathrm{s}$ and a long path is representative for movements between all reachable poses in the workspace of the manipulator. If $T$ is not set, the desired terminal pose $H^e_{w,\mathrm{des}}$, will be approached by minimizing the objective function within the NMPC, just bounded to the given OCP constraints.

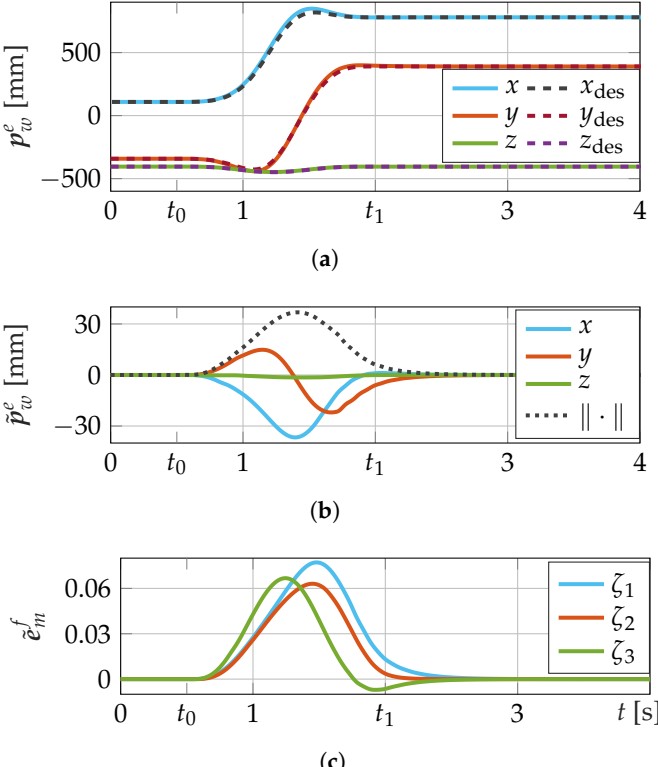

**Figure 3.** Trajectory of the end-effector starting at $t_0 = 0.5\,\mathrm{s}$ and ending at $t_1 = 2\,\mathrm{s}$ in an undisturbed environment for tracking accuracy analysis in 3D space. (**a**) Trajectory tracking of the pre-planned trajectory marked by the subscript "des", which is planned in the joint space and transformed into the Cartesian space. (**b**) Absolute displacement of the end-effector $\tilde{p}^e_w = p^e_{w,\mathrm{des}} - p^e_w$ to the reference trajectory and the individual translational parts depending on the length $d_e = 150\,\mathrm{mm}$ of the end-effector. (**c**) Orientation error (11) in quaternion representation.

The pre-planned trajectory is generated by using the polynomial depicted in (7) in the joint space. The computation time of approximately 1 ms required for this involving the automatic joint configuration selection ensures an almost instantaneous start. Subsequently, this is transformed to the Cartesian space. As can be seen in Figure 3a, the reference trajectory exhibits rounded deviations, for example at $t = 1.2\,\mathrm{s}$ in $p^e_{w,z,\mathrm{des}}$, compared to a trajectory that would be directly planned in the Cartesian space, because the joints are actuated uniformly over $T$ here. In turn, the evaluation of the orientation by, e.g., roll-pitch-yaw [74] is omitted by using the joint space, which could be singular in the representation.

In the evaluation of the end-effector's translational deviation, both controllers must be taken into account. It should be noted that the end-effector position is composed of the positioning of the wrist point by the NMPC and the alignment by the orientation controller. Both a too slow control of the wrist point and an incorrect orientation of the end-effector would lead to a deviation from the end-effector's reference trajectory. In Figure 3b, the individual error components of $\tilde{p}^e_w = p^e_{w,\mathrm{des}} - p^e_w$ and the absolute distance $\|\tilde{p}^e_w\|$ to the reference trajectory at each time step are shown. Despite the short transition time $T$ and the long displacement along the trajectory, resulting in a rapid change of poses, only small deviations can be detected. Compared to a common path tracking task, it must be taken into account that, in the analysis of the trajectory tracking accuracy, a slight lag also leads

to notable deviations. As can be seen in Figure 3b, especially the errors of $p^e_{w,x}$ and $p^e_{w,y}$ exhibit small deviations, which converge to zero in the end, so that no stationary error remains. The small lag in the $x_w y_w$-plane during the motion results from the parameterized smoothness of the orientation controller, since it must perform the rotation correction in each iteration step due to its constantly shifting reference $\{m\}$ system. The NMPC places the wrist point $p^\varsigma_w$ very accurately so that the reference system of the upper kinematic chain, used for the orientation control, is moved further and further by the NMPC. Therefore, a permanent adaptation in (11) governed by $K$ is necessary. The rotation error is shown in Figure 3c, where each of the imaginary unit quaternion error components can take a maximum value of one. Thus, it can be seen that the orientation error was very small in this case. Even though, the end-effector is chosen to be relatively long with $d_e = 150$ mm. As a result, a larger deviation was enforced for a better illustration here. If $d_e$ is chosen shorter, the amplitudes in Figure 3b decrease. In total, just small deviations from the pre-planned trajectory and, thus, accurate trajectory tracking can be observed when using the presented approach.

*4.3. Trajectory Control in Disturbed Environment With Fixed Obstacles*

In the evaluation involving obstacles, the same control setup as in Section 4.2 was utilized, but as illustrated in Figure 2, the scenario now included $\nu = 4$ obstacles and $p^e_{w,0} = [110, -350, -445]^\mathsf{T}$ mm and $p^e_{w,\text{des}} = [780, 390, -445]^\mathsf{T}$ mm were set 40 mm lower in the $z_w$-direction. This small lowering of the reference trajectory would cause ground contact, which should be prevented by the controller. Starting from the blue marker in Figure 4, the first obstacle was placed close to the reference trajectory so that the boundary condition (10f) had to consider the mentioned safety distance, since the NMPC has no knowledge about the orientation controller, which adjusts the desired orientation. In the extreme case, when the end-effector would be vertical, the NMPC should directly leave the reference trajectory to avoid collisions. As shown in Figure 4, the two consecutive obstacles on the left-hand side are crossed by the blue reference and disturb the tracking of the pre-planned trajectory in $x_w y_w$-plane. Additionally, the central obstacle ($I_\nu$) presents a difficulty in conjunction with the height constraint (10g), since the NMPC must deviate significantly from the reference and take a remarkable detour to reach $H^e_{w,\text{des}}$. The NMPC only considers the obstacles within the prediction horizon and has no information about them before. In the accompanying video [92], the orientation error and the wrist point tracking are also shown, in addition to the executed robot movements. For the sake of readability, the evaluation is omitted in this section and reference is made to Section 4.2.

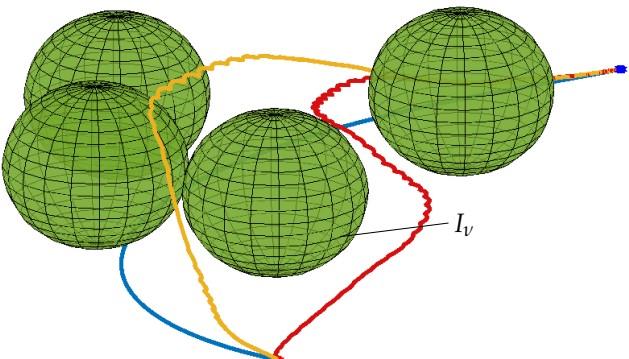

**Figure 4.** Resulting trajectories of the end-effector $p^e_w(t)$ governed by the NMPC and orientation controller in the scenario from Figure 2. The motion starts at the blue marker and ends at the red marker. The reference trajectory (blue) according to (7) crosses the obstacles (green). Without a height constraint for $p^\varsigma_{w,z}$, the motion results in an upward swerve (yellow). Activating (10g), the spheres are avoided in the $x_w y_w$-plane (red).

From Figure 5a, the trajectory of the end-effector can be taken in the case where the wrist point $p^\varsigma_w$ is only constrained by $p^\varsigma_{w,z,\text{min}} = -228$ mm in (10g) involving no upper

height limit, so that the ground will not be touched. A deviation from the dashed reference trajectory due to the obstacle avoidance can be seen. Especially with respect to $p_{w,x}^e$ and $p_{w,y}^e$, it is obvious that the trajectory controller tries to follow the reference trajectory under consideration of the given constraints, but a delay is recognizable. From Figure 4, it becomes even clearer that the yellow trajectory in the $x_w y_w$-plane follows the arc of the blue reference quiet accurately. The obstacles are avoided by swerving in the $z_w$-direction, which is confirmed by the green line in Figure 5a. Due to the chosen IPOPT algorithm and depending on the length of the control horizon, which has to be chosen according to the computation times of the controller's online calculations, small repeated repulsions of the end-effector can be detected in Figure 4 while avoiding the obstacles tightly.

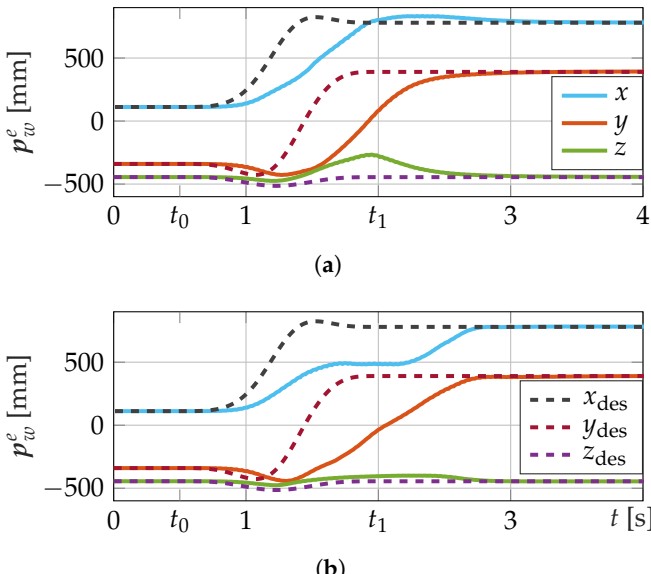

**Figure 5.** Trajectory tracking of the blue curve in Figure 4 starting at $t_0 = 0.5\,\text{s}$ and ending at $t_1 = 2\,\text{s}$ in a disturbed environment involving $\nu = 4$ fixed obstacles. (**a**) No upper height limitation of the Cartesian space. Analogous to the yellow trajectory in Figure 4, the manipulator moves over the obstacles. (**b**) Constraining the height by $-228\,\text{mm} \leq p_{w,z}^\zeta \leq -145\,\text{mm}$ in (10g) for obstacle avoidance in the $x_w y_w$-plane, as the red curve in Figure 4.

By reducing the computational costs, short evaluation times of the NMPC can be achieved, even if additional constraints are inserted, which further influence the robot's behavior. This means that, by decomposing the differential kinematics, not only an accurate controller can be designed, but also, it can be used more flexibly. In order to demonstrate this, the maximum height in the Cartesian space was constrained in the further analysis. The height constraint $-228\,\text{mm} \leq p_{w,z}^\zeta \leq -145\,\text{mm}$ of the wrist's workspace forces the controller to avoid the obstacles by a planar motion. The lifting of the end-effector is thus suppressed. Therefore, $H_{w,\text{des}}^e$ can just be approached by a significant deviation from the reference trajectory, mainly disturbed by the central obstacle ($I_\nu$). A noticeable change in the movement compared to the dashed lines can be noticed at Figure 5b. Even though, the motion has to be adapted and is thus slightly delayed. The online applicability of the approach is still valid. For a better interpretation, the corresponding course is illustrated as the red path in Figure 4. This shows that the introduced approach is able to control the standard industrial robot in disturbed environments.

*4.4. Trajectory Control in a Varying Environment with Moving Obstacles*

Based on the evaluation of the controller in a disturbed environment, the same setup as shown in Figures 2 and 4 with $p_{w,0}^e = [110, -350, -445]^\text{T}\text{mm}$ and $p_{w,\text{des}}^e = [780, 390, -445]^\text{T}\text{mm}$ was utilized subsequently. The Jacobian transpose controller continued to align the end-effector downward. However, the $\nu = 4$ obstacles were

in motion here, and thus, they represent a varying environment. Again, the wrist point was constrained in height using $-228\,\text{mm} \le p^C_{w,z} \le -145\,\text{mm}$ to avoid an upward swerve. Two different scenarios were examined to demonstrate the resulting behavior of the controller. First, all obstacles moved uniformly in one direction, continuously blocking the corridor realized Figure 5b after the end-effector deviated from the reference trajectory. Subsequently, only the central obstacle $I_v$ moved, which caused a dead end for a short time. Both scenarios were chosen such that the obstacles force the NMPC to adjust the movement and the end-effector must depart from the desired trajectory analogous to Section 4.3. The reference trajectory shown in Figure 5 and, thus, the motion of the robot starts at $t_0 = 0.5\,\text{s}$ and ends at $t_1 = 2\,\text{s}$.

A uniform movement of the obstacles simulated the driving of the robot, as is common, e.g., in agricultural or industrial applications. Starting at $t = 0\,\text{s}$, the four obstacles moved uniformly with $30\,\text{mm}\,\text{s}^{-1}$ in the negative $x_w$-direction and with $45\,\text{mm}\,\text{s}^{-1}$ in the $y_w$-direction, so that they moved diagonally towards the robot. The $\{w\}$ system can be taken from Figure 2, which is aligned with the axes of the $\{0\}$ frame. The change in position of the moving obstacles relative to the yellow robot can be seen in Figure 6a–g. The safe distance introduced in (10f) is illustrated in orange. The trajectory of the online controlled movement can be taken from Figure 7a. For comparison, the resulting trajectories from the previous subsection are also plotted in Figure 7 and marked with the subscript "fix". Due to the height constraint of the wrist in the NMPC, any adjustment of the movement in the $z_w$-direction was excluded. However, compared to the static scenario, further adjustments were conducted in the $x_w y_w$-plane. The end-effector was also guided into the corridor between the obstacles after passing the first one from the robot's point of view, as the rear ones prevent the direct tracking of the trajectory.

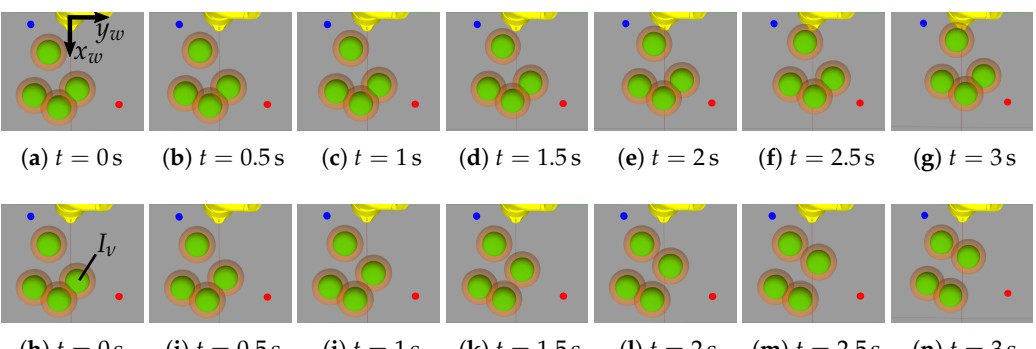

**Figure 6.** The positions of the moving obstacles (green) at different time steps. The safe distance (orange) from (10f) and the robot (yellow) are highlighted for clarification. Referring to the $\{w\}$ system in Figure 2, (**a**–**g**) illustrate the uniform movement of all obstacles with $30\,\text{mm}\,\text{s}^{-1}$ in the negative $x_w$-direction and with $45\,\text{mm}\,\text{s}^{-1}$ in the $y_w$-direction. In (**h**–**n**), the displacement of the central obstacle $I_v$ with $60\,\text{mm}\,\text{s}^{-1}$ in the negative $x_w$-direction is visualized.

Due to the displacement of the obstacles, the end-effector can be shifted earlier in the $y_w$-direction, which can be observed from the comparison of the curves visualizing $p^e_{w,y}$. At about $t = 1.7\,\text{s}$, the NMPC must further adjust the motion to avoid a collision with the central obstacle marked by $I_v$ in Figure 2 and Figure 4, respectively. Therefore, the end-effector is initially pulled closer to the base, which is apparent from $p^e_{w,x}$, and then, a fast movement is performed to pass $I_v$. The increase in velocity can be seen in Figure 7a at about $t = 2.4\,\text{s}$ by the larger slope of $p^e_{w,x}$ and $p^e_{w,y}$. The desired terminal pose is reached insignificantly later without collision.

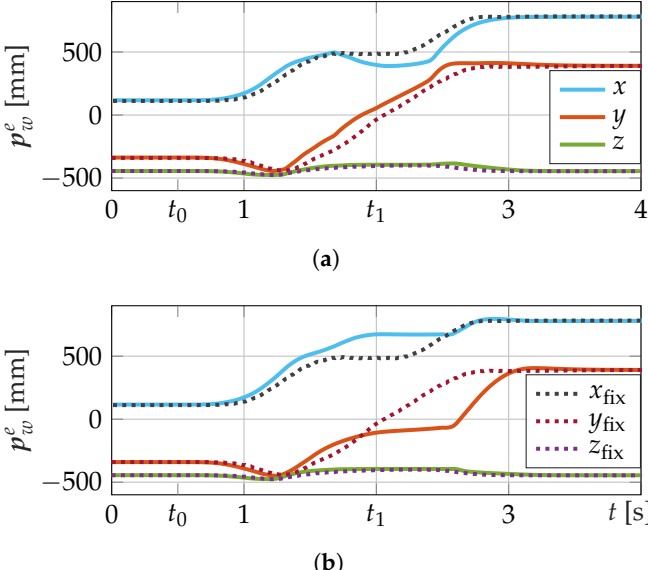

**Figure 7.** Trajectory tracking of the blue curve in Figure 4 starting at $t_0 = 0.5\,\text{s}$ and ending at $t_1 = 2\,\text{s}$ in a disturbed environment involving moving obstacles and an active height constraint. The directions of the obstacles' movement can be taken from Figure 2, and for comparison, the subscript "fix" indicates the end-effector's motion in the statically disturbed environment from Figure 5b. (**a**) All obstacles move uniformly with $30\,\text{mm\,s}^{-1}$ in the negative $x_w$-direction and with $45\,\text{mm\,s}^{-1}$ in the $y_w$-direction. (**b**) The central obstacle ($I_v$) moves with $60\,\text{mm\,s}^{-1}$ in the negative $x_w$-direction, so that a dead end is created briefly.

In the second scenario evaluating the decomposition-based controller in a varying environment, only the central obstacle ($I_v$) moved with $60\,\text{mm\,s}^{-1}$ in the negative $x_w$-direction, and the others were fixed. Figure 6h–n demonstrate the movement of $I_v$. The corresponding trajectory of the end-effector is shown in Figure 7b. It can be seen that the beginning of the movement was similar to the motion performed in the static disturbed environment. The NMPC departed from the blue reference trajectory in Figure 4 to move through the corridor between the obstacles. However, the movement of $I_v$ was chosen in such a way that this briefly formed a dead end in combination with the other obstacles, while the end-effector tried to pass it. This led to a deceleration of the movement between approximately $t = 1.4\,\text{s}$ and $t = 1.9\,\text{s}$, since $\boldsymbol{p}^e_{w,\text{des}}$ cannot be reached at that moment. Due to the round shape of the obstacles, a small adjustment of the motion can be detected from that time in $p^e_{w,y}$ as $I_v$ continued to move on. From about $t = 2.6\,\text{s}$, the corridor between $I_v$ and the rear obstacles opened enough to continue the motion without any collisions, so that the desired terminal pose was reached. This demonstrates that the presented optimal control approach can be used in disturbed and varying environments.

## 5. Discussion

In this paper an approach that reduces the computational costs of NMPC was introduced and applied for online trajectory control in disturbed and varying environments. For this purpose, the differential kinematics was decomposed and partitioned into a translational and rotational part related to the Cartesian space containing the corresponding robot joints. The differential kinematics was considered since it can be modeled based on the DH parameters from the robot's data sheets, and thus, the approach is also applicable to position controlled industrial robots. The decomposition-based approach can be applied to all robot types that can be partitioned in this way. The translational motion control was used for obstacle avoidance. For this purpose, an OCP was introduced and implemented as the NMPC, which considered both Cartesian and joint constraints. Due to the reduced number of decision variables in the OCP, additional constraints to adjust the robot's behavior, such as the height of the workspace, can be included without significantly

increasing the computation times. The NMPC moved the wrist point of the anthropomorphic robot, which led to a change in the orientation of the end-effector. For the correction of the orientation, the Jacobian transpose controller was introduced and applied to the problem using unit quaternions, avoiding singularities. In addition to the online motion control, trajectory planning with an automatic selection of the best joint configuration was introduced, eliminating the need for manual input, as usually required for PTP motions.

In the evaluation, the approach was compared in terms of the computational costs with an NMPC that considered the full robot model involving all DOFs. The comparison demonstrated that the required times in the computation of the OCP were significantly reduced by the introduced method. For the analysis, different scenarios and parameters were considered. A comparison with other approaches has been omitted here, since higher computation times resulted from the references presented in the introduction, e.g., in [23–25,48], and the consideration of the full system is most common.

The analysis of the tracking accuracy and the delay-free online control in a disturbed or varying environment with (moving) obstacles was performed in simulations. The implementation was carried out using a standard computer, MATLAB, ROS, and CASADI, although the performance in hardware and software can be further increased by replacing these tools. Larger prediction horizons and differently shaped obstacles can also be considered. However, this work serves as a proof of concept and was intended to show the possibilities of this decomposition-based approach. The extensive evaluation revealed that the coupled controllers precisely followed a trajectory and adapted the motion to the environment. This resulted in an optimal controller setup that considered external constraints with high precision and without limiting the workspace. The focus was placed on the NMPC, although the Jacobian transpose controller was also considered in the evaluation, but this has been extensively analyzed in other publications [58,69]. Further, it was examined whether the orientation was also implemented by a second NMPC whose initial conditions were given by the translational NMPC.

This contribution introduced the concept of the control architecture and evaluated it by simulations. This demonstrated the precision of the method and serves as a basis for further developments. Besides the replacement of MATLAB by another programming language, improvements will be made to the individual components for the transfer to a real experiment. ROS is suitable for the communication with GAZEBO, but for a better performance, this will be replaced by ETHERCAT [93], as, e.g., implemented in [94] for the control of a Stäubli TX2-90. The obstacles are detected by motion tracking, and their positions can be updated in (10f) at a high rate. Alternative solvers such as ACADOS [95,96] or GraMPC [97] are evaluated either on a Linux server or on a PLC to further reduce the computation times.

## 6. Conclusions and Future Work

This contribution exhibited that, by decomposing the differential kinematics of an anthropomorphic robot, the computational costs of NMPC can be significantly reduced with basically no effect on the solution's accuracy and reliability. This seemingly small adjustment has a huge impact on the computational effort and demonstrates an approach that addresses the cause, not the symptoms, of long NMPC computation times. By reducing the decision variables in the OCP, optimal online trajectory control in disturbed and varying environments for (standard industrial) robots is possible. The NMPC for the translational motion of the end-effector was coupled with a Jacobian transpose controller for the orientation correction, so that all DOFs of the robot were used. There were no special requirements for the control hardware, and a standard computer was sufficient for the NMPC evaluations. The simulation results showed that an online implementation for NMPC in the field of robotics has been elaborated without limiting the workspace due to the model's decomposition. This opens up the possibility of using standard industrial robots in various areas and applications, where many sensor data have to be processed or the interaction with a dynamically varying environment is required. The evaluation

of the computation times, the tracking accuracy, the control in a disturbed environment with additional height constraints, and the trajectory adaptation in a varying environment demonstrated the performance of the approach.

The method offers many possibilities in terms of extension and transferability. In addition to self-collision avoidance, the concept can also be used to interact with objects due to the short evaluation times of the NMPC. Further on, the approach will be brought to a real experiment and coupled with a force control, so that, e.g., haptic grasping can be implemented in a disturbed environment.

**Author Contributions:** Conceptualization, J.R., H.B. and T.M.; methodology, J.R.; software, J.R. (decomposed system and evaluation) and H.B. (full system for comparison); validation, J.R. and H.B.; formal analysis, J.R. and H.B.; investigation, J.R. and H.B.; resources, T.M.; data curation, J.R.; writing—original draft preparation, J.R.; writing—review and editing, J.R., H.B. and T.M.; visualization, J.R.; supervision, J.R. and T.M.; project administration, J.R. and T.M.; funding acquisition, T.M. All authors have read and agreed to the published version of the manuscript.

**Funding:** The authors gratefully acknowledge the financial support by the federal state of Schleswig-Holstein within the funding programme Open Access Publikationsfonds. Furthermore, this research received no external funding.

**Data Availability Statement:** The data presented in this study are available upon request from the corresponding author.

**Conflicts of Interest:** The authors declare no conflict of interest.

## Abbreviations

The following abbreviations are used in this manuscript:

| | |
|---|---|
| 3D | three-dimensional |
| des | desired |
| DH | Denavit–Hartenberg |
| DOF | degree of freedom |
| e.g., | for example |
| Fig. | Figure |
| fix | fixed |
| IPOPT | interior-point |
| NMPC | nonlinear model predictive control |
| OCP | optimal control problem |
| Tab. | Table |

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
