# Peer review of "Constrained-Differential-Kinematics-Decomposition-Based NMPC for Online Manipulator Control with Low Computational Costs"

_robotics, doi:10.3390/robotics12010007_

Round 1

Reviewer 1 Report

This paper deals with a practical challenge when it comes to online implementation of nonlinear MPC algorithm, namely, how to deal with computational burden. There are many methods that have been developed over decades to deal with this challenge, and most of them are either related to treating the computational delay as input/output delay or to have sample based MPC. However, in this work, the authors have targeted a very specific way to handle computational burden associated with MPC, and their method is confined to controlling robot motion. So this technique of handling computational delay is specific application oriented, and not a general method, and the authors have mentioned about this in the paper.

Overall, a well written paper which deserves to be published.

Reviewer 2 Report

This manuscript proposed Online Robot Control for Low Computational Costs using Nonlinear Model Predictive Control and Constrained Differential Kinematics Decomposition.

 My detailed comments are listed as below:

1-     It is not clear what the contribution is relative to the state of the art. I think the authors should identify more specifically what it is they propose that is novel and interesting, and with respect to what specific existing works it is novel and interesting. This information is not clearly presented in this manuscript.

2-     The proposed approach is supported by theoretical analysis and modelling and there is not any sufficient experimental validation or comparing the results with other research.

3-     In section 3, the authors claimed that they did Optimal Trajectory Control Using Decomposed Differential Kinematics. But there is not clear what is objective for optimization. If it is computational costs to solve online the optimal control problem, then how did they optimize it? And it is not clear how the equation 10b shows the computational costs?

4-     In section 4, obstacle – end effector collision is addressed, and it is worthwhile if they can cover the robot arms- obstacle collision.

In these conditions I can suggest major revision for the present work.

Reviewer 3 Report

The author proposed an approach that analyzes and decomposes the differential kinematics similar to the inverse kinematics method to assign Cartesian boundary conditions during the model building to reduce the online computational cost. The proposed method is validated in the simulated environment using GAZEBO 14 using a Stäubli TX2-90.
Overall, the paper is well written and clear in content, the objective of the paper is stated clearly and the paper is executed in a proper way throughout the paper. However the content can be improved for the readability and correctness following points are suggested.
Comments:
1.    In the introduction author claimed, “The proposed method is suitable for the application in various fields including industrial robots,” can author add some applications in the introduction?
2.    In the introduction the computational cost is presented. The author may refer and cite (say at in Page 2 first paragraph) one existing paper for the inverse kinematics using different approaches as in “Robot manipulation through inverse kinematics” with ACM where it is reported with the wrist partitioned robot.
Also the advantage of the proposed approach can be highlighted with the other state of the art methods as highlighted in “Reinforcement learning based compensation methods for robot manipulators”.
3.    In section 4, what is the rationale behind choosing interior-point (IPOPT) algorithm for optimization?
4.    In section 4.2 author discussed about the trajectory tracking accuracy of the introduced approach, can author add some results about the NMPC based full system and compare the trajectory tracking accuracy between these two? If possible, please compare the results in the section 4.3 and 4.4 in a similar fashion to further validate your approach.
5.    To further support the claim made in the section 4, can author add additional timestamp image (Position at different time intervals) for Trajectory Control in Varying Environment with Moving Obstacles.
6.    The error in Fig. 3b seems to be high upto 3cms. The error in the orientation is not highlighted. The reason for it should be discussed in the discussion and the respective results section.
7.    Authors are suggested to highlight the results with some standard trajectory like in cartesian space with constant orientation and indicate the time for the comparison purpose.
Minor comments:
Authors are suggested to relook and rewrite the following claims and sentences:
(Line 74 Pg 2: If only the Cartesian space is considered, the OCP neglects all nonlinearities of the robot model and does not take the reachable work and joint space into account [44,45].), (Line 82: Obstacle avoidance in 3D space is primarily done by translational movements),
Line 200: Element of R>0 is not clear?
Equation 12: R^(3 x3), should be in R^3 only. And the size of the matrix is $3 \times 3$
Line 271 iterations as done, e.g., in [81]. E.g can be deleted

Reviewer 4 Report

This paper presents a method for control of manipulator robots based on NMPC with low computational cost, which is achieved by using a decomposition of the differential kinematics of the robot. The results clearly show the benefits of the proposed approach.

Since the approach is not valid for all kind of robots, I suggest to change the paper title as: Constrained Differential Kinematics Decomposition Based NMPC for Online Manipulator Robot Control with Low Computational Cost

The paper is in general very well written, there is only some few typos, but the paper is easy to reading. However, I have the following comments that must be taken into account:

1) The abstract is too long, the first 7 lines are not really necessary. Write the abstract in a compact and concise form. 

2) Clarify in the introduction, line 114 whether the polynomial planned trajectory is generated online or it is only planned at the beginning.

3) Above eq. (1): ...reduce the number of variables in the equations [20,61]. Specify which equations are the authors referring.

4) Page 4, line 145, introduce that a method to select one of the eight possible solutions is proposed later.

5) Make clearer how the entire workspace is still reachable using the decoupled kinematics, as concluded in line 158.

6) Introduce in line 182 that the orientation controller is based on Transpose Jacobian.

7) Describe some of the ideas of references [14,21,23,66] and [20,69,70] cited in page 8, because they are referred without no details.

8) At the end of Section 3.2, comment about the feasibility and convexity of the OCP. Moreover, explain what happens if no solution is found in the OCP.

9) About Table 2, explain how lines 4 and 5 must be interpreted, since one expect to see the table of DH parameters with only 6 lines.

10) In line 266, specify the value of the control cycle or update rate of the NMPC. Also explain what are the grid points and their effect in the method. Relate to the same aspect, explain the information in lines 302-303, in particular about the discretization steps.

11) In line 271, what does it mean “until an optimal solution was found”? Clarify the stop condition.

12) Include a video of the simulation with moving obstacles to make clear how they move.

13) Discuss what is needed to implement the proposed scheme in real experiments.

14) Revise the manuscript for typos and grammar issues: configuraion, wirst, are not be violated, Lapunov, what happens, Therefor.

Round 2

Reviewer 2 Report

I am satisfied with this author response.